# The Association between Cervical Length and Successful Labor Induction: A Retrospective Cohort Study

**DOI:** 10.3390/ijerph20021138

**Published:** 2023-01-09

**Authors:** Pei-Chen Li, Wing Lam Tsui, Dah-Ching Ding

**Affiliations:** 1Department of Obstetrics and Gynecology, Hualien Tzu Chi Hospital, Buddhist Tzu Chi Medical Foundation, Tzu Chi University, Hualien 970, Taiwan; 2Institute of Medical Sciences, Tzu Chi University, Hualien 970, Taiwan

**Keywords:** labor induction, cervical length, Bishop score, ultrasound, vaginal delivery, cesarean delivery

## Abstract

This study aimed to determine whether transvaginal sonographic measurement of cervical length before labor induction can predict successful induction. This retrospective study recruited 138 pregnant women who underwent labor induction at 37–41 weeks of gestation. Cervical length was measured using transvaginal ultrasonography before labor induction. Labor was induced according to the hospital protocol. Age, gestational age (GA), parity, body mass index (BMI), Bishop score, hemoglobin level, maternal disease, and epidural anesthesia were also recorded. Labor induction outcomes, including cesarean section for failed induction, time of induction, and the three labor stages, were assessed. From December 2018 to December 2021, 138 women were recruited for our study, including 120 and 18 women with successful and failed labor induction, respectively. Shorter cervical length (≤3.415 cm, OR = 6.22, 95% CI = 1.75–22.15) and multiparity (OR = 17.69, 95% CI = 2.94–106.51) were associated with successful induction. Higher BMI was associated with failed induction (OR = 0.87, 95% CI = 0.75–0.99). Age, GA, Bishop score, and fetal birth weight were not associated with successful labor induction. The ROC curve showed a cervical length cutoff value of 3.415 cm, revealing 76.8% of the area under the curve. In conclusion, a shorter cervical length (≤3.415 cm) was associated with a higher chance of successful labor induction (76.8%). This parameter might be used to predict the chance of successful labor induction. This information could help better inform clinician discussions with pregnant women concerning the chance of successful labor induction and consequent decision-making. Nevertheless, further large-scale clinical trials are warranted.

## 1. Introduction

The need for induction of labor (IOL) has risen over recent decades and is required in approximately 25% of term pregnancies in the United States [1]. The rate of IOL was 18.4% in nullipara and 10.2% in multipara in China from 2015 to 2016 [2]. A previous study published in 2007 has shown IOL was associated with an increased cesarean section (C/S) rate [3]. However, a recent study in 2019, the ARRIVE trial, showed IOL nullipara without risk factors at 39 weeks of gestation reduced the cesarean section (C/S) rate [4]. An increasing amount of evidence from meta-analyses suggests that IOL at 39 weeks is associated with a lower risk of C/S, maternal peripartum infection, and adverse perinatal outcomes [5,6].

The decision to undergo labor induction has several advantages. It can reduce the risk of macrosomia and its consequences, such as shoulder dystocia or the need for instrumental delivery, which may lead to maternal trauma [7]. However, it can also decrease the risk of adverse neonatal outcomes, such as meconium aspiration syndrome and neonatal death, as studies have shown that these events can occur at 40 weeks of gestation [8]. However, elective labor induction may also lead to adverse outcomes. Some studies have proposed that mechanical or pharmacological induction may increase the risk of cesarean section [9]. Elective labor also increases medical costs for induction media compared to a natural birth course.

Cervical ripening and uterine contraction are important for labor induction. The current medication used for both uterine activities is a prostaglandin product. Approval for IOL medication is prostaglandin E2. Two formulae are available in Taiwan, prostin (dinoprostone vaginal tablet) and propess (dinoprostone vaginal pessary). In our previous study, propess and prostin had similar efficacy for IOL in primipara [10].

The Bishop score is a useful tool for predicting cervical maturity and determining whether induction is successful. During a vaginal exam, obstetricians gather information on cervical effacement, consistency, dilation, position, and fetal station, which are the five critical components of the Bishop score. Each item is rated from zero to two or three points. A score below four indicates that the cervix is not ripped, and induction should not start without the use of cervical ripening agents. A score of six or above is more favorable, and induction is more likely to succeed [11].

Despite the wide application of the Bishop scoring system, some recent studies have questioned its reliability, as the process of evaluation may sometimes be subjective and different depending on each physician’s skills and experience [12,13]. Hence, newer studies have proposed that measuring cervical length may be a more accurate and objective means to determining whether induction will be successful [14]. It is considered a new tool for obstetricians despite the uncertainties regarding the use of this novel measurement in evaluating the success rate of induction. Therefore, this study aimed to evaluate the association between cervical length and successful labor induction.

## 2. Materials and Methods

### 2.1. Study Design

This retrospective cohort study included women with singleton pregnancies who underwent labor induction at 37–41 weeks of gestation between December 2018 and December 2021 (Figure 1).

### 2.2. Inclusion and Exclusion Criteria

Women with singleton pregnancies with vertex presentation who underwent labor induction at 37–41 weeks of gestation were included. Women with spontaneous labor, preterm delivery, prelabor rupture of membranes, history of previous cervical surgeries, multiple pregnancies, malpresentation, previous uterine scar, and a Bishop score of more than 8 at the time of hospital admission were excluded from the study. A Bishop score of more than 8 was excluded because the chance of delivery with induction is similar to that of spontaneous labor [15].

### 2.3. Induction Protocol

The pregnant woman was admitted to our ward, and the fetal heart rate was monitored for 30 min. A vaginal examination was performed, and induction medication was inserted simultaneously. The induction medications included prostin (prostaglandin E2 vaginal tablet), propess (prostaglandin E2 intravaginal slow-release device), and Piton (oxytocin); each can be used alone or in combination. We usually prescribed propess for nullipara and prostin for multipara. Propess could be retained in the vagina for 24 h. The prostin tablet could be inserted every 4 h if there was a lack of labor progress. Oxytocin could be used for augmentation of labor if a rupture of membranes after induction was noted. The combination use depended on the progress of labor. If there was no progress in labor, a second medication would be added. Due to propess being a self-paid medication, if patients could not afford it, it was not prescribed. Prostin and propess were used intravaginally. Piton was administered intravenously. Fetal monitor tracing noted labor induction in at least four contractions with at least 70 amplitudes at 20 min intervals. The vaginal medication was stopped when the following condition occurred: nonreassuring fetal heart rate, rupture of the membranes, or uterine tachysystole.

### 2.4. Outcome Measurement

Preinduction cervical length was measured using transvaginal ultrasonography during the study period. Due to cervical length being an important factor for successful labor induction, we routinely measured cervical length during the study period. Cervical length measurement is not routine in most clinics. The linear distance from the internal os to the external os of the cervix was measured [16].

### 2.5. Primary Outcome

Successful induction was defined as vaginal delivery within 72 h of labor induction. After 72 h, two conditions would be encountered. One was keeping labor (prolonged latent phase), and the other was C/S due to obstetric reasons (prolonged labor, nonreassuring fetal status).

### 2.6. Other Characteristics Collected from the Patient

Other characteristics were also collected, including age, body mass index (BMI), hemoglobin (Hb) level, parity, group B streptococcus (GBS) status, indication, induction method, use of epidural anesthesia, Bishop score, total induction time, stages of labor, delivery mode, and birth weight.

### 2.7. Statistical Analysis

Categorical variables were measured using the Chi-squared test or Fisher’s exact test. Continuous variables were measured using an independent t-test or Mann–Whitney U test. Multivariate analysis was performed using the logistic and linear regression models. ROC (receiver operating characteristic) curve and cutoff value were calculated. Statistical analysis was performed using SPSS software (version 25, IBM, Chicago, IL, USA). A *p*-value < 0.05 was considered statistically significant.

## 3. Results

### 3.1. Demographics

Table 1 shows the demographic characteristics of the study population, which consisted of 138 women who underwent labor induction [successful induction, n = 120 (86.9%); failed induction, n = 18 (13.1%)]. Cervical length was shorter in the successful induction than in the failed induction group (2.84 ± 0.76 cm vs. 3.52 ± 0.65 cm, *p* < 0.001). Lower BMI (28.74 ± 3.93 kg/m^2^ vs. 30.92 ± 5.15 kg/m^2^, *p* = 0.037) and more multiparity (52.5% vs. 88.9%, *p* = 0.004) were noted in the successful induction group. Indications for labor induction were significantly different between the two groups (*p* = 0.016). More oligohydramnios and gestational hypertension were present in the failed labor induction group. The total induction time was shorter in the successful induction group than in the failed induction group (1132.23 ± 800.95 min vs. 5194.50 ± 446.18 min, *p* < 0.001). The Bishop score was not significantly different between the two groups (2.90 ± 2.21 cm vs. 2.00 ± 2.22 cm in the successful and failed induction groups, respectively, *p* = 0.110).

### 3.2. Factors Associated with Successful Induction

Table 2 shows factors associated with successful induction. Shorter cervical length (≤3.415 cm, OR = 6.22, 95% CI = 1.75–22.15) and multiparity (OR = 17.69, 95% CI = 2.94–106.51) were associated with successful induction. Higher BMI was associated with failed induction (OR = 0.87, 95% CI = 0.75–0.99). Age, GA, Bishop score, and fetal birth weight were not associated with successful labor induction after adjustment.

### 3.3. The Percentage of Successful Labor Induction in Different Cervical Lengths

Table 3 shows the percentage of successful induction in different cervical length groups. In the cervical length <3 cm group, successful induction reached 98.6%. However, in the cervical length >4 cm group, successful induction reduced to 60.0%.

### 3.4. Factors Associated with Total Labor Time

Table 4 shows factors associated with total labor time. After the covariables shown in Table 1 were adjusted, shorter cervical length (*p* = 0.005) and multiparity (*p* < 0.001) was associated with a significantly short labor time (*p* = 0.005). Age, BMI, GA, Bishop score, and fetal birth weight were not associated with total labor time after adjustment.

### 3.5. ROC Curves of Cervical Length and Successful Induction

The ROC curves for the optimal cutoff value of cervical length are shown in Figure 2. The ROC curve showed the tradeoff between specificity and sensitivity. The sensitivity was 0.775, and the specificity was 0.611. The curve closer to the top-left corner indicates better performance. The cutoff value for cervical length was 3.415 (area under the curve: 0.768).

The glossary of terms is explained in the Glossary.

## 4. Discussion

This retrospective cohort study concluded that in women who delivered vaginally after elective induction, the cervix length was shorter than that in women with failed induction. Multiparity was also associated with successful induction. We also found that a higher BMI significantly lengthens labor time.

With the rising prevalence of obesity among pregnant women, statistics have shown that obesity and labor induction have a dose-dependent relationship [17]. This relationship can be explained by the fact that when a woman’s BMI is high, the risk of pregnancy complications, such as gestational diabetes, gestational hypertension, and macrosomia, increases. When these complications develop, extra labor induction is often required according to their severity [17]. Lundborg et al. observed a longer active first stage of labor in obese women [18]. With the increased risk of prolonged labor, Palatnik et al. found that women with a higher BMI should plan elective induction programs, thereby decreasing the need for emergent cesarean delivery [19]. Erberla et al. also found in their retrospective cohort study that women with higher BMI who underwent induction of labor at 39 weeks of gestation had a lower risk of cesarean delivery than those who did not [20]. A higher BMI certainly lengthens labor time, so elective induction should be carefully planned in women with obesity.

Despite the possible higher sensitivity in predicting whether induction will be successful, the time of application of cervical length measurement is still being debated. Most of the related studies have involved cervical measurements before induction [21]. This prevents the cervical characteristics from being changed by the prostaglandin gel used for induction [22].

Abdullah et al. discovered that preinduction cervical length measurement was an independent predictive factor of whether induction would be successful [23]. It was mentioned in their study that a cervical length cutoff of 27 mm would be favorable for successful induction. However, the results of some studies have contrasted with our study results; Khandelwal et al. found that preinduction cervical length measurement was not associated with the induction-to-delivery success rate [24]. The differences in results may be due to the different definitions of successful induction, as some studies defined successful induction as delivery within 24 h. Meanwhile, the proficiency of the use of transvaginal ultrasound by each independent examiner may contribute to the difference in results, as a measurement of cervical length at term pregnancy tends to be more difficult as the fetus’s head engages.

In this study, we found that a shorter cervical length (≤3.415 cm) was associated with a higher chance of successful labor induction (76.8%). As the need for elective induction increases, further investigation of other prediction models is of utmost importance. Recent studies have shed light on the possibility of using elastography as a predictive tool. Wang et al. reported promising accuracy in using shear wave elastography (SWE) to measure cervical elastography [25]. In addition to cervical elastography, some studies have suggested that biomarkers, such as fetal fibronectin and insulin-like growth factor binding protein-1, may help predict whether induction will be successful. Grab et al. suggested with their study that both cervical length and fetal fibronectin are independent predictors of successful induction [26]. Further investigations are required to examine the accuracy and application of these potential predictive tools.

Multiparity is a favorable factor for IOL. In one study of IOL with prostaglandin E2, multiparous women benefited more than did primiparous ones [27]. Daykan et al. also reported IOL with propess was more successful in multiparous women than in primiparous ones [28]. A randomized controlled trial that recruited 200 term pregnant women who received prostin and propess also found parity to be associated with successful IOL [29]. In line with these studies, our study also found multiparity to be associated with a higher chance of successful IOL.

### Strengths and Limitations

In this study, we used multivariate regression analysis to identify the factors associated with successful induction. Hence, we could control for a few potential cofounders while accessing each independent relationship. As this was a retrospective cohort study within a hospital setting, we could also gather a complete patient profile through medical records, so there was no recall bias. However, our study had a small sample size and used different induction protocols. We defined successful induction as vaginal delivery within 72 h, while other studies usually define it as delivery within 24 h. This might have led to a disagreement in the results when compared to those of other studies. Cervical length was not the only factor associated with failed labor induction, which was also the limitation of this study. As we did not carry out formulation studies involving the height, pelvic entrance, and pubic arch measurements of the patients, achieving definitive clinical results with these data is limited. Physical activity or exercise may affect the delivery time, but they were not recorded in this study. Finally, prenatal ultrasonographic estimation of fetal weight measurements and transition times from the latent phase to the active phase were not evaluated together with labor induction.

## 5. Conclusions

A shorter cervical length (≤3.415 cm) was associated with a higher chance of successful labor induction (76.8%). This parameter might be used to predict the chance of successful labor induction. This information could better inform clinician discussion with pregnant women regarding their chance of successful labor induction and in the consequent step in decision-making. Nevertheless, further large-scale clinical trials are warranted.

## Figures and Tables

**Figure 1 ijerph-20-01138-f001:**
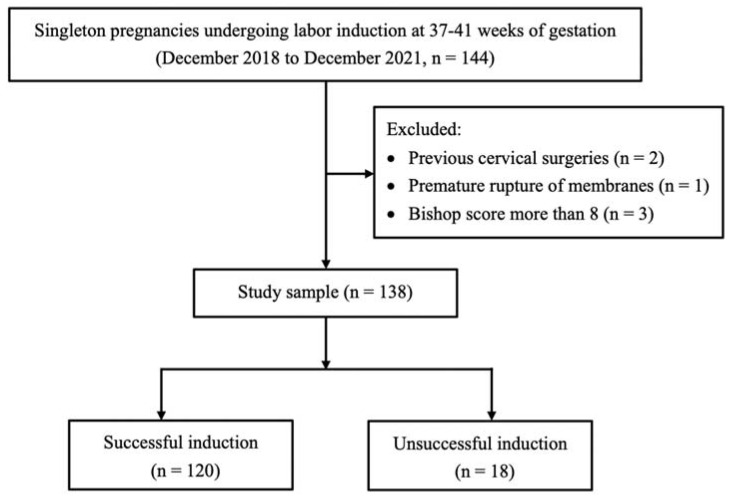
Study flowchart.

**Figure 2 ijerph-20-01138-f002:**
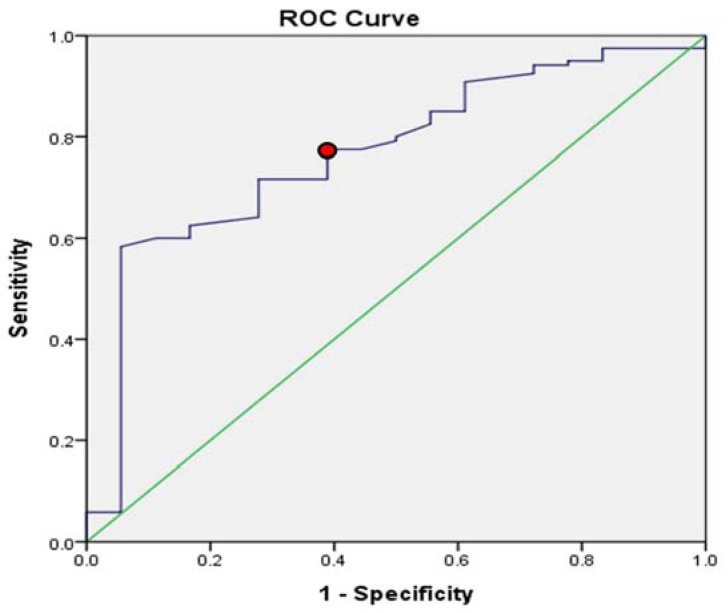
ROC curve for cervical length and successful induction. The ROC curve showed a cervical length set at 3.415 cm (area under the curve: 0.768); the sensitivity was 0.775, and the specificity was 0.611.

**Table 1 ijerph-20-01138-t001:** Demographic characteristics of the study population (n = 138).

	Successful Induction		Total	*p*-Value
	No	Yes		
N (%)	18 (13.1%)	120 (86.9%)	138	
Age (years)	31.56 ± 5.74	29.98 ± 5.48	30.19 ± 5.52	0.262
Weight (kg)	77.52 ± 15.05	74.20 ± 9.77	74.63 ± 10.60	0.216
BMI (kg/m^2^)	30.92 ± 5.15	28.74 ± 3.93	29.02 ± 4.16	0.037 *
GA (weeks)	38.75 ± 0.73	39.05 ± 0.87	39.01 ± 0.85	0.174
GA group (weeks)				0.790
37 + 0~37 + 6	2 (11.1%)	8 (6.7%)	10 (7.2%)	
38 + 0~38 + 6	9 (50%)	49 (40.8%)	58 (42.0%)	
39 + 0~39 + 6	5 (27.8%)	44 (36.7%)	49 (35.5%)	
40 + 0~40 + 6	2 (11.1%)	17 (14.2%)	19 (13.8%)	
41 + 0~41 + 6	0 (0%)	2 (1.7%)	2 (1.4%)	
Hb (g/dL)	11.27 ± 1.39	10.93 ± 1.54	10.97 ± 1.52	0.370
Parity [n (%)]	-	-	-	0.004 *
Nulliparity	16 (88.9%)	63 (52.5%)	79 (57.2%)	
Multiparity	2 (11.1%)	57 (47.5%)	59 (42.8%)	
Maternal disease [n (%)]	-	-	-	
Normal	14 (77.8%)	90 (75.0%)	104 (75.4%)	1.000
IDA	1 (5.6%)	20 (16.7)	21 (15.2%)	0.308
Autoimmune disease	0 (0.0%)	3 (2.5%)	3 (2.2%)	1.000
Asthma	1 (5.6%)	2 (1.7%)	3 (2.2%)	0.345
Vitamin D deficiency	0 (0.0%)	2 (1.7%)	2 (1.4%)	1.000
Thyroid disease	0 (0.0%)	2 (1.7%)	2 (1.4%)	1.000
Cancer	0 (0.0%)	2 (1.7%)	2 (1.4%)	1.000
Thalassemia	2 (11.1%)	0 (0.0%)	2 (1.4%)	0.016 *
HBV	0 (0.0%)	1 (0.8%)	1 (0.7%)	1.000
Sicca syndrome	0 (0.0%)	1 (0.8%)	1 (0.7%)	1.000
Prenatal problem [n (%)]	-	-	-	0.055
Normal	12 (66.7%)	98 (81.7%)	110 (79.7%)	
GDM	2 (11.1%)	11 (9.2%)	13 (9.4%)	
IUGR	1 (5.6%)	5 (4.2%)	6 (4.3%)	
Gestational HTN	0 (0.0%)	2 (1.7%)	2 (1.4%)	
Polyhydramnios	1 (5.6%)	1 (0.8%)	2 (1.4%)	
Preeclampsia	2 (11.1%)	0 (0.0%)	2 (1.4%)	
SUA	0 (0.0%)	2 (1.7%)	2 (1.4%)	
Fetal arrhythmia	0 (0.0%)	1 (0.8%)	1 (0.7%)	
GBS (%)	4 (22.2%)	22 (18.3%)	26 (18.8%)	0.747
Indication of labor induction [n (%)]	-	-	-	0.016 *
No labor sign	13 (72.2%)	110 (91.7%)	123 (89.1%)	
IUGR	1 (5.6%)	5 (4.2%)	6 (4.3%)	
Oligohydramnios	2 (11.1%)	4 (3.3%)	6 (4.3%)	
Gestational HTN/Polyhydramnios/Preeclampsia	2 (11.1%)	1 (0.8%)	3 (2.2%)	
Induction method [n (%)]	-	-	-	0.155
Prostin	5 (27.8%)	54 (45.0%)	59 (42.8%)	
Propess	4 (22.2%)	34 (28.3%)	38 (27.5%)	
Propess + Piton	2 (11.1%)	13 (10.8%)	15 (10.9%)	
Prostin + Piton	3 (16.7%)	9 (7.5%)	12 (8.7%)	
Propess + Prostin	2 (11.1%)	3 (2.5%)	5 (3.6%)	
Propess + Prostin + Piton	2 (11.1%)	7 (5.8%)	9 (6.5%)	
Epidural anesthesia [n (%)]	11 (61.1%)	86 (71.7%)	97 (70.3%)	0.361
Cervical length (cm)	3.52 ± 0.65	2.84 ± 0.76	2.93 ± 0.78	<0.001 *
Cervical length group				<0.001 *
<3.0 cm	1 (5.6%)	70 (58.3%)	71 (51.4%)	
3.0–3.5 cm	8 (44.4%)	26 (21.7%)	34 (24.6%)	
>3.5–4 cm	5 (27.8%)	18 (15.0%)	23 (16.7%)	
>4 cm	4 (22.2%)	6 (5.0%)	10 (7.2%)	
Funneling [n (%)]	2 (11.1%)	29 (24.2%)	31 (22.5%)	0.363
Bishop score	2.00 ± 2.22	2.90 ± 2.21	2.78 ± 2.23	0.110
Position [n (%)]	-	-	-	0.698
Posterior	13 (72.2%)	93 (77.5%)	106 (76.8%)	
Mid	5 (27.8%)	24 (20.0%)	29 (21.0%)	
Anterior	0 (0.0%)	3 (2.5%)	3 (2.2%)	
Dilation [n (%)]	-	-	-	0.499
Closed	14 (77.8%)	79 (65.8%)	93 (67.4%)	
1–2 cm	4 (22.2%)	40 (33.3%)	44 (31.9%)	
3–4 cm	0 (0.0%)	1 (0.8%)	1 (0.7%)	
Effacement [n (%)]	-	-	-	1.000
0–30%	15 (83.3%)	93 (77.5%)	108 (78.3%)	
40–50%	3 (16.7%)	23 (19.2%)	26 (18.8%)	
60–70%	0 (0.0%)	4 (3.3%)	4 (2.9%)	
Consistency [n (%)]	-	-	-	0.127
Firm	10 (55.6%)	36 (30.0%)	46 (33.3%)	
Medium	0 (0.0%)	1 (0.8%)	1 (0.7%)	
Soft	8 (44.4%)	83 (69.2%)	91 (65.9%)	
Station [n (%)]	-	-	-	0.683
−3	9 (50.0%)	47 (39.2%)	56(40.6%)	
−2	8 (44.4%)	65 (54.2%)	73(52.9%)	
−1, 0	1 (5.6%)	8 (6.7%)	9(6.5%)	
Induction time (min)	5194.50 ± 446.18	1132.23 ± 800.95	1198.82 ± 949.13	<0.001 *
Stages of labor (min)	-	-	-	
1st stage	1150.50 ± 1448.86	325.17 ± 404.26	338.70 ± 434.91	0.007 *
2nd stage	54.50 ± 21.92	78.35 ± 112.95	77.96 ± 112.07	0.767
3rd stage	9.00 ± 2.83	4.71 ± 5.43	4.78 ± 5.42	0.268
Delivery mode [n (%)]	-	-	-	<0.001 *
NSD	1 (5.6%)	85 (70.8%)	86 (62.3%)	
VED	1 (5.6%)	35 (29.2%)	36 (26.1%)	
C/S	16 (88.9%)	0 (0.0%)	16 (11.6%)	
Birth weight (g)	3115.61 ± 348.53	3130.88 ± 327.08	3128.88 ± 328.67	0.855
Sex of fetus [n (%)]	-	-	-	0.643
Female	9 (50.0%)	67 (55.8%)	76 (55.1%)	
Male	9 (50.0%)	53 (44.2%)	62 (44.9%)	

Data are presented as n (%) or mean ± standard deviation. * *p*-value < 0.05 was considered statistically significant after testing. BMI—body mass index; GA—gestational age; Hb—hemoglobin; IDA—iron deficiency anemia; HBV—hepatitis B virus; GDM—gestational diabetes mellitus; IUGR—intrauterine growth restriction; HTN—hypertension; SUA—single umbilical artery; GBS—group B streptococcus; NSD—normal spontaneous delivery; VED—vacuum extraction delivery; C/S—cesarean section; min—minutes.

**Table 2 ijerph-20-01138-t002:** Factors associated with successful induction (n = 138).

	Crude		Adjusted	
	Odds Ratio (95% CI)	*p*-Value	Odds Ratio (95% CI)	*p*-Value
Age (years)	0.95 (0.86, 1.04)	0.262	0.94 (0.84, 1.06)	0.320
BMI (kg/m^2^)	0.89 (0.79, 0.99)	0.042 *	0.87 (0.75, 0.99)	0.044 *
GA (weeks)	1.53 (0.83, 2.83)	0.174	2.13 (0.95, 4.78)	0.066
Bishop score	1.22 (0.95, 1.56)	0.114	0.92 (0.69, 1.23)	0.878
Cervical length	-	-	-	-
>3.415 cm	Reference (1.0)		Reference (1.0)	
≤3.415 cm	5.41 (1.91, 15.31)	0.001 *	6.22 (1.75, 22.15)	0.005 *
Parity	-	-	-	-
Nulliparity	References		References	
Multiparity	7.24 (1.59, 32.86)	0.010 *	17.69 (2.94, 106.51)	0.002 *
Fetal birth weight (kg)	1.15 (0.26, 5.15)	0.854	0.63 (0.10, 4.03)	0.629

Data are presented as odds ratio (95% CI); * *p*-value < 0.05 was considered statistically significant after testing. BMI—body mass index; GA—gestational age.

**Table 3 ijerph-20-01138-t003:** Cervical length vs. successful induction (n = 138).

	N	Successful Induction		*p*-Value
		No	Yes	
Cervical length group				<0.001 *
<3.0 cm	71	1 (1.4%)	70 (98.6%)	
3.0–3.5 cm	34	8 (23.5%)	26 (76.5%)	
>3.5–4.0 cm	23	5 (21.7%)	18 (78.3%)	
>4 cm	10	4 (40.0%)	6 (60.0%)	
Total	138	18 (13.0%)	120 (87.0%)	

* *p*-value < 0.05 was considered statistically significant after testing.

**Table 4 ijerph-20-01138-t004:** Factors associated with total labor time (n = 138).

	Crude			Adjusted		
	Regression Coefficient	95% CI	*p*-Value	Regression Coefficient	95% CI	*p*-Value
Age (years)	10.92	(−20.52, 42.36)	0.493	8.68	(−18.98, 36.35)	0.535
BMI (kg/m^2^)	6.50	(−36.81, 49.80)	0.767	21.61	(−15.31, 58.53)	0.249
GA (weeks)	41.18	(−158.52, 240.89)	0.684	−26.55	(−210.92, 157.83)	0.776
Bishop score	−114.41	(−189.13, −39.69)	0.003 *	−50.10	(−121.74, 21.55)	0.169
Cervical length (cm)	300.82	(84.26, 517.38)	0.007 *	301.87	(92.82, 510.92)	0.005 *
Parity	-	-	-	-	-	-
Nulliparity	Reference	Reference	NA	Reference	Reference	NA
Multiparity	−873.95	(−1177.76, −570.13)	<0.001 *	−947.74	(−1256.94, −638.53)	<0.001 *
Fetal birth weight (kg)	47.26	(-479.98, 574.50)	0.859	231.55	(−220.45, 683.54)	0.312

Dependent variable: total labor time. * *p*-value < 0.05 was considered statistically significant after the testing. GA—gestational age; BMI—body mass index.

## Data Availability

All data were included in the manuscript.

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
