# Peer review of "The Association between Cervical Length and Successful Labor Induction: A Retrospective Cohort Study"

_ijerph, 2023, doi:10.3390/ijerph20021138_

Round 1
Reviewer 1 Report
I would like to thank all the authors for this difficult study design and their efforts. Your study is well-prepared and fluent, but a few appendices are required for study limitations;
1- Type of induction of labor (written as a hospital protocol, but should be standardized and stated in the article, we applied it in the same dose and time in each patient + was it administered vaginally? Was oxytocin administered intravenously? Which criterion was observed for labor induction (at least 20 minutes in NST tracing such as 4 contractions with at least 70 amplitude..) should be added to the method section, such as when was it stopped?
2- It would be nice if we add a sentence similar to the sentence 'As we did not carry out formulation studies involving the height, pelvic entrance and pubal arch measurements of the patients, achieving definitive clinical results is limited with these data' to the study limitations.
3- Since we do not know clearly the physical activity or exercises (yoga, pilates, etc.) that patients do individually or professionally for pelvic relaxation during pregnancy follow-up, it is difficult to confirm the cross-relationship between delivery time and cervical length.
4- Since parameters such as prenatal ultrasonographic estimation of fetal weight measurements and transition times from latent phase to active phase are not evaluated together with labor induction, the inclusion of these parameters in future studies may provide more precise data. If a sentence like ' is added to the limitations, it will look more elegant.
Yours truly
Author Response
I would like to thank all the authors for this difficult study design and their efforts. Your study is well-prepared and fluent, but a few appendices are required for study limitations;
1- Type of induction of labor (written as a hospital protocol, but should be standardized and stated in the article, we applied it in the same dose and time in each patient + was it administered vaginally? Was oxytocin administered intravenously? Which criterion was observed for labor induction (at least 20 minutes in NST tracing such as 4 contractions with at least 70 amplitude..) should be added to the method section, such as when was it stopped?
Response: We added induction protocol in the methods section 2.3.
2- It would be nice if we add a sentence similar to the sentence 'As we did not carry out formulation studies involving the height, pelvic entrance and pubal arch measurements of the patients, achieving definitive clinical results is limited with these data' to the study limitations.
Response: We added the sentence to the limitation section.
3- Since we do not know clearly the physical activity or exercises (yoga, pilates, etc.) that patients do individually or professionally for pelvic relaxation during pregnancy follow-up, it is difficult to confirm the cross-relationship between delivery time and cervical length.
Response: We added this point to the limitation section.
4- Since parameters such as prenatal ultrasonographic estimation of fetal weight measurements and transition times from latent phase to active phase are not evaluated together with labor induction, the inclusion of these parameters in future studies may provide more precise data. If a sentence like ' is added to the limitations, it will look more elegant.
Response: We added this point to the limitation section.
Reviewer 2 Report
Reviewer statement:
Transvaginal Cervical Length Measurement to Predict Successful Labor Induction,
Reviewer statement:
Induction of labor is a very common intervention in obstetrics. The need for induction has increased over recent decades. Induction seems to reduce the risk of cesarean delivery and adverse perinatal outcomes but may also lead to adverse outcomes. The method of induction is crucial but more important from a clinical view is the prediction of cervical maturity and determining whether induction is going to be successful. As noticed by the authors, there is still a lot debate if the cervical length may be a more accurate and objective mean to determine whether induction will be successful. The authors report the result of a retrospective, control study which is very interesting and a clinical relevant topic.
Title:
The title is misleading, as this not reflect the study as presented. The authors should adjust the title, to reflect the study being presented. Furthermore, I would suggest to add a retrospective study to the title.
Overall:
The paper is well written and attractive to read from a reader point of view. The aim of the study is well emphasized and explained in the manuscript. From a readers perspective it was a pleasure to read.
Abstract
The abstract section is well written and appropriate but should be adjusted on basis on provided comments.
Introduction
The introduction section is clear, emphasizing the current knowledge and the reasons for conduction this retrospective study.
No comments on this section.
Materials and Methods
This section is well written, explaining the method the study was conducted. A couple of points require explanation and or clarification.
1. Women with a Bishop score of more than 8 were excluded from the study. Form a reader point of view it is not clear and logic to exclude these patients as they are very prone to deliver spontaneously and the length of their cervix is also interesting to know. Please elucidate om this point and clarify to the reader the reason for exclusion.
2. The authors report on page 2, line 71: “Pre-induction cervical length was measured using transvaginal ultrasonography.” Could the authors why the cervical length was routinely measured in their clinic as this not routinely performed in daily clinical practice? The authors should explain this important fact to the readers.
3. Furthermore, the question remains what was done with the result of the cervical length. Did the result have an impact on the provided treatment?
4. The authors included women who underwent labor induction between 37–41 weeks of gestation. As the authors report themselves in the introduction section elective labor induction at 39 weeks is associated with improved outcomes. Please elucidate on this important point, as successful induction is dependent on gestation age.
5. Successful induction was defined as vaginal delivery within 72 h of labor induction. What was done at 72 hours and how this registered and handle in the provided results: this should be reported, please do so?
Results
This section is well written, explaining the method the study was conducted. A couple of points require explanation and or clarification.
6. The aim of the study was to evaluate the association between cervical length and successful labor induction. The authors should report the results in a fashion to try to answer the aim. The fashion the result are presented by the authors does not answer the aim of the study.
7. From a readers point of view and for daily practice one starts with a cervical length, and want you want to know what the chance is of a successful induction,. The authors should report the results primary in this fashion. please do so?
8. The authors divided the group in success full and failed induction, while the reason for a failed induction is not only the cervical length, the authors should acknowledge this and this is an important limitation.
9. When looking at table 1 important and crucial points require explanations. The method of induction is different. When was prostin used and/ or propess? The authors should elucidate thoroughly on this important point.
10. Furthermore, the indication for labor, parity are significant different between the two groups, the authors should report and elucidate on these point.
11. Although there was no significant difference in the Bishop score, the mean bishop score of the failed induction group was 0.9 lower, this should also be acknowledge, we do not know if this has an impact on the presented result.
12. The authors choose a cervical length of 3.415 as a cut of point, on basis of which facts. This should be explained thoroughly to the reader. Please do so.
Discussion
This section is well written, explaining the method the study was conducted. A couple of points require explanation and or clarification.
13. The authors conclude: “This retrospective cohort study concluded that longer cervical length leads to a higher chance of induction failure and prolonged labor time. “ The conclusion is incorrect as the authors did not examine cervix length in relation to the chance of successful induction. On basis of the presented result the conclusion should be: in women who delivered vaginally after elective induction the cervix length was shorter that in women with failed induction.” This also implies for BMI.
14. The authors also report: parity is negatively associated with labor time, in agreement with long-standing consensus. As the authors report themselves, this is long-standing consensus; the authors should have corrected for this important influencing factor. When looking at table 2, there is significant difference between the groups on this crucial point, making interpretation of the presented result almost impossible. The authors should acknowledge this fact.
15. Moreover, the sentence in line 156 should be adjusted : “ In this study, we concluded that pre-induction measurement of cervical length is useful for predicting whether induction would be successful.” On basis of the provided result, this conclusion is not eligible. Please adjust.
16. The authors should conclude that on basis of the represented result firm conclusions are not possible, as there are several important influencing factors, such as parity, term of induction and BMI which was possible to correct for: important limitations of the study are not reported sufficiently and/ or corrected for.
Conclusion
17. Should be adjusted on basis of earlier remarks.
Figure
No comments.
Author Response
Induction of labor is a very common intervention in obstetrics. The need for induction has increased over recent decades. Induction seems to reduce the risk of cesarean delivery and adverse perinatal outcomes but may also lead to adverse outcomes. The method of induction is crucial but more important from a clinical view is the prediction of cervical maturity and determining whether induction is going to be successful. As noticed by the authors, there is still a lot debate if the cervical length may be a more accurate and objective mean to determine whether induction will be successful. The authors report the result of a retrospective, control study which is very interesting and a clinical relevant topic.
Title:
The title is misleading, as this not reflect the study as presented. The authors should adjust the title, to reflect the study being presented. Furthermore, I would suggest to add a retrospective study to the title.
Response: We changed the title to: “The Association Between Cervical Length and Successful Labor Induction: a Retrospective Cohort Study”.
Overall:
The paper is well written and attractive to read from a reader point of view. The aim of the study is well emphasized and explained in the manuscript. From a readers perspective it was a pleasure to read.
Response: Thank you for the comment.
Abstract
The abstract section is well written and appropriate but should be adjusted on basis on provided comments.
Response: We revised the abstract thoroughly.
Introduction
The introduction section is clear, emphasizing the current knowledge and the reasons for conduction this retrospective study.
No comments on this section.
Materials and Methods
This section is well written, explaining the method the study was conducted. A couple of points require explanation and or clarification.
- Women with a Bishop score of more than 8 were excluded from the study. Form a reader point of view it is not clear and logic to exclude these patients as they are very prone to deliver spontaneously and the length of their cervix is also interesting to know. Please elucidate om this point and clarify to the reader the reason for exclusion.
Response: We added a reference to elucidate it “due to the chance of delivery with induction is similar to spontaneous labor (Wormer et al. 2022).”
- The authors report on page 2, line 71: “Pre-induction cervical length was measured using transvaginal ultrasonography.” Could the authors why the cervical length was routinely measured in their clinic as this not routinely performed in daily clinical practice? The authors should explain this important fact to the readers.
Response: We want to know the influence of cervical length on successful labor induction. Therefore, we routinely measure the cervical length on admission during the study period. But this is not routine in most clinics. We added this point in the methods section 2.4.
- Furthermore, the question remains what was done with the result of the cervical length. Did the result have an impact on the provided treatment?
Response: After measurement of cervical length, we can provide the chance of successful labor induction to the pregnant woman. Maybe it can succeed, but it may take a long time. If the chance of successful labor induction is low, C/S could be considered. We added this point to the conclusion section.
- The authors included women who underwent labor induction between 37–41 weeks of gestation. As the authors report themselves in the introduction section elective labor induction at 39 weeks is associated with improved outcomes. Please elucidate on this important point, as successful induction is dependent on gestation age.
Response: We added gestational age into table 2 and 3. The analysis showed no significant influence on successful labor induction.
- Successful induction was defined as vaginal delivery within 72 h of labor induction. What was done at 72 hours and how this registered and handle in the provided results: this should be reported, please do so?
Response: After 72 hours, two conditions will be encountered. One is keeping labor (prolonged latent phase). Another is C/S due to obstetric reasons (prolonged labor, non-reassuring fetal status). We added this point into methods section 2.5.
Results
This section is well written, explaining the method the study was conducted. A couple of points require explanation and or clarification.
- The aim of the study was to evaluate the association between cervical length and successful labor induction. The authors should report the results in a fashion to try to answer the aim. The fashion the result are presented by the authors does not answer the aim of the study.
Response: We rewrote the results section thoroughly.
- From a readers point of view and for daily practice one starts with a cervical length, and want you want to know what the chance is of a successful induction,. The authors should report the results primary in this fashion. please do so?
Response: We rewrote the results section with successful induction fashion.
- The authors divided the group in success full and failed induction, while the reason for a failed induction is not only the cervical length, the authors should acknowledge this and this is an important limitation.
Response: We added this limitation in the limitation section.
- When looking at table 1 important and crucial points require explanations. The method of induction is different. When was prostin used and/ or propess? The authors should elucidate thoroughly on this important point.
Response: We added this point in the methods section (2.3. induction protocol).
- Furthermore, the indication for labor, parity are significant different between the two groups, the authors should report and elucidate on these point.
Response: We added this point in the results.
- Although there was no significant difference in the Bishop score, the mean bishop score of the failed induction group was 0.9 lower, this should also be acknowledge, we do not know if this has an impact on the presented result.
Response: We added this point in the results.
- The authors choose a cervical length of 3.415 as a cut of point, on basis of which facts. This should be explained thoroughly to the reader. Please do so.
Response: We added this point in the results.
Discussion
This section is well written, explaining the method the study was conducted. A couple of points require explanation and or clarification.
- The authors conclude: “This retrospective cohort study concluded that longer cervical length leads to a higher chance of induction failure and prolonged labor time. “ The conclusion is incorrect as the authors did not examine cervix length in relation to the chance of successful induction. On basis of the presented result the conclusion should be: in women who delivered vaginally after elective induction the cervix length was shorter that in women with failed induction.” This also implies for BMI.
Response: We revised the sentence accordingly.
- The authors also report: parity is negatively associated with labor time, in agreement with long-standing consensus. As the authors report themselves, this is long-standing consensus; the authors should have corrected for this important influencing factor. When looking at table 2, there is significant difference between the groups on this crucial point, making interpretation of the presented result almost impossible. The authors should acknowledge this fact.
Response: We added this variable in table 2.
- Moreover, the sentence in line 156 should be adjusted : “ In this study, we concluded that pre-induction measurement of cervical length is useful for predicting whether induction would be successful.” On basis of the provided result, this conclusion is not eligible. Please adjust.
Response: We adjusted the sentence to “In this study, we concluded that a shorter cervical length (≤ 3.415 cm) was associated with a higher chance of successful labor induction (76.8%).”
- The authors should conclude that on basis of the represented result firm conclusions are not possible, as there are several important influencing factors, such as parity, term of induction and BMI which was possible to correct for: important limitations of the study are not reported sufficiently and/ or corrected for.
Response: We added that cervical length is not the only factor associated with failed labor induction in the limitation section.
Reviewer 3 Report
The topic in general is interesting and may be attractive to readers, however to be published in this journal I note that both the introduction, methodology and conclusions are too brief for this high impact journal. I suggest the authors to add more content to the introduction.
Author Response
The topic in general is interesting and may be attractive to readers, however to be published in this journal I note that both the introduction, methodology and conclusions are too brief for this high impact journal. I suggest the authors to add more content to the introduction.
Response: We added one paragraph to the introduction section and lengthen the methods and conclusion.
Reviewer 4 Report
I am grateful for the possibility to review this article which is related to the success rates of induction of labour.
The overall assessment of the article is that from the introduction to the proposed topic, its state of the art seems to be scarce.
Commonly known variables are evaluated, although it is true that they provide important data which have already been dealt with.
Within the discussion line 157-158: it seems that we find ourselves in a conclusion and not a discussion, which is poor in the presentation of articles for that, to discuss the subject, a matter that would improve the research. Furthermore, in this section, it is recommended to end with proposals to be carried out after the research carried out, in order to improve and apply in the process of induced labour.
Author Response
I am grateful for the possibility to review this article which is related to the success rates of induction of labour.
The overall assessment of the article is that from the introduction to the proposed topic, its state of the art seems to be scarce.
Commonly known variables are evaluated, although it is true that they provide important data which have already been dealt with.
Within the discussion line 157-158: it seems that we find ourselves in a conclusion and not a discussion, which is poor in the presentation of articles for that, to discuss the subject, a matter that would improve the research. Furthermore, in this section, it is recommended to end with proposals to be carried out after the research carried out, in order to improve and apply in the process of induced labour.
Response: We rewrote the sentence. We also rewrote the conclusion to provide clinical application for labor induction.
Reviewer 5 Report
The manuscript "Transvaginal Cervical Length Measurement to Predict Successful Labor Induction" is an interesting manuscript on the role of transvaginal cervical length measurement to predict successful labor induction. The work is not so original, but it is well-performed and structured. The design of the project is appropriate and the results are significant. The statistical analysis is well conducted and the language is acceptable. What are the actual clinical implications of this study? it is important to report the results obtained by the authors in the context of clinical practice and to adequately highlight what contribution this study adds to the literature already existing on the topic and to future study perspectives.
Author Response
The manuscript "Transvaginal Cervical Length Measurement to Predict Successful Labor Induction" is an interesting manuscript on the role of transvaginal cervical length measurement to predict successful labor induction. The work is not so original, but it is well-performed and structured. The design of the project is appropriate and the results are significant. The statistical analysis is well conducted and the language is acceptable.
What are the actual clinical implications of this study? it is important to report the results obtained by the authors in the context of clinical practice and to adequately highlight what contribution this study adds to the literature already existing on the topic and to future study perspectives.
Response: We rewrote the conclusion to provide clinical application for labor induction. We also expanded introduction and discussion section.
Round 2
Reviewer 2 Report
The authors have revised the manuscript and reply on the remarks, suggestions and comments of the reviewers and have significantly improved the quality of the article. I complement the authors for this achievement. Although, the curent article is suitable to be published, the auhtors could imporve the clinical significance of this article by added/ adjusted some remaining points, hereby increasing the excellence of this artcile.
Before reporting the remaining pints, I want to congratulate the authors with is achievement and this is a important and very relevant clinical topic.
1. The aim of the study as declared by the authors themselves was to evaluate the association between cervical length and successful labor induction. From a clinical point of view you want to know if the cervix length is e.a. between 3.5- 4.0 cm how great is the chance that the induction will be successful? And so on for the different cervix lengths. On basis of the available data, this should be possible to provide and this is much more relevant to clinicians than the way the authors report the result now.
2. The authors report the mean gestation age in tables 2 and 3 . Although this is excellent, this is incomplete data for interpretation to the reader. It remains unknow how many women were inducted at 37+0- to 37-6, 38+0 to 38+6 in the groups and so. The mean gestation does not provide sufficient information, the authors could provide this in a supplemental table.
3. It remains remarkable, that of the 120 women with successful induction, no one of the women received a cesarean section. This needs explanation.
4. In section 2.3 , the authors report a lot of new influences factors, increasing the chance of bias.
5. Piton is introduced as a new alternative medication, not reported earlier.
6. In the Strength and limitation section the authors declare: “ Hence, we could control for potential cofounders while accessing each independent relationship.” Based on the revised manuscript we can concluded that this facts is not justified. Although, the authors tried to control potential cofounders, they could not have adjusted for all influencing factors, please adjust this sentence.
Author Response
Reviewer 2
The authors have revised the manuscript and reply on the remarks, suggestions and comments of the reviewers and have significantly improved the quality of the article. I complement the authors for this achievement. Although, the curent article is suitable to be published, the auhtors could imporve the clinical significance of this article by added/ adjusted some remaining points, hereby increasing the excellence of this artcile.
Before reporting the remaining pints, I want to congratulate the authors with is achievement and this is a important and very relevant clinical topic.
- The aim of the study as declared by the authors themselves was to evaluate the association between cervical length and successful labor induction. From a clinical point of view you want to know if the cervix length is e.a. between 3.5- 4.0 cm how great is the chance that the induction will be successful? And so on for the different cervix lengths. On basis of the available data, this should be possible to provide and this is much more relevant to clinicians than the way the authors report the result now.
Response: We added this point to the result (new Table 3).
- The authors report the mean gestation age in tables 2 and 3 . Although this is excellent, this is incomplete data for interpretation to the reader. It remains unknow how many women were inducted at 37+0- to 37-6, 38+0 to 38+6 in the groups and so. The mean gestation does not provide sufficient information, the authors could provide this in a supplemental table.
Response: We added these data in table 1.
- It remains remarkable, that of the 120 women with successful induction, no one of the women received a cesarean section. This needs explanation.
Response: The C/S cases were counted as a failure of labor induction. Therefore, cases in successful group contained no C/S cases. The C/S rate in this study was 11.6% which was the same as the rate in our hospital.
- In section 2.3 , the authors report a lot of new influences factors, increasing the chance of bias.
Response: The induction protocol was originally designed. The influence factors were already counted in.
- Piton is introduced as a new alternative medication, not reported earlier.
Response: Piton had already listed in Table 1 as propess+piton, prostin+piton, or propess+prostin+piton. We changed the description regarding piton.”Oxytocin was used for augmentation of labor if ruptured of membranes after induction were noted”
- In the Strength and limitation section the authors declare: “ Hence, we could control for potential cofounders while accessing each independent relationship.” Based on the revised manuscript we can concluded that this facts is not justified. Although, the authors tried to control potential cofounders, they could not have adjusted for all influencing factors, please adjust this sentence.
Response: We revised this sentence to “Hence, we could control for a few potential cofounders while accessing each independent relationship.”
Reviewer 3 Report
Aceptar el artículo.
Author Response
Aceptar el artículo.
Response: Thanks for the comments.
Reviewer 4 Report
In this case, the article has improved substantially with the contributions made by following the recommendations of the reviewers.
A fair conclusion is objective.
It is recommended to include a glossary of terms used for a better understanding of the research.
Author Response
In this case, the article has improved substantially with the contributions made by following the recommendations of the reviewers.
A fair conclusion is objective.
It is recommended to include a glossary of terms used for a better understanding of the research.
Response: We added a glossary of terms in supplemental table 1.